# Killing of colistin-resistant *Aeromonas hydrophila* by a synthetic peptide

Leisheng Sun,[1,2] Zonghan Jiang,[3] Dingding Li,[1] Yao Tian,[4] Junqi Liu,[4] Zhiliang Sun[3]

**ABSTRACT**  Multi-drug-resistant *Aeromonas hydrophila* infections are becoming increasingly threatening, and the development of novel antimicrobial drugs is indispensable. Herein, we demonstrate that this novel peptide is highly active against colistin-resistant *A. hydrophila* strain and shows sustained killing efficacy *in vivo*. Mechanistic studies showed that D-Q7 interacted with phosphatidylglycerol and lipopolysaccharide in the bacterial cell membrane, with an increase in intracellular ROS as well as a decrease in ATP level, ultimately leading to cell membrane disruption and bacterial death. Importantly, our study identified gene3832 as a potential regulator of membrane permeability, which may act as a potential modulator of bacterial susceptibility to D-Q7. The role of gene3832 was further confirmed by gene knockout and complementation assays. Consistently, we observed that gene3832 was also involved in biofilm formation in the colistin-resistant *A. hydrophila* strain. Collectively, our study provides an effective antimicrobial strategy with potential targets for the treatment of drug-resistant *A. hydrophila* infection.

**IMPORTANCE**  As an environmental, zoonotic pathogen, *Aeromonas hydrophila* remains a major pathogenic bacterium, bringing large economic losses and eco-environmental pressure during the event of large-scale infection. Currently, the occurrence of colistin-resistant *A. hydrophila* poses a threat to public health owing to the lack of effective prevention and therapeutics. D-Q7 is a D-type antimicrobial peptide (AMP) with potent sterilization activity against gram-negative ESKAPE pathogens; it is thus of considerable interest to evaluate whether D-Q7 represents a promising therapeutic candidate against this pathogen. Consequently, we found that D-Q7 was a potent antibacterial agent that killed colistin-resistant *A. hydrophila 23-c-23 in vitro* and in a mouse epicutaneous model of *23-c-23* infection. In addition, we found that *gene3832*, as a potential transmembrane autotransporter, is related to bacterial resistance to D-Q7. Importantly, our study here will help guide the future design and optimization of novel AMPs to combat colistin-resistant *A. hydrophila*.

**KEYWORDS**  *Aeromonas hydrophila*, antimicrobial agents, drug resistance

*A*eromonas hydrophila is characterized as one of the main pathogens of fish and other marine animals, leading to notorious outbreaks in aquaculture systems because of their wide distribution in aquatic environments (1). Besides, pathological processes generated by these opportunistic bacteria are frequently reported in immunosuppressed patients. Several virulence factors in this pathogen, including lipases, proteases, hemolysins, aerolysins, cytotoxins, and enterotoxins, are shown to play roles in the survival and environmental adaptation as well as infectious processes (2). Due to the overuse of various antibiotics, *A. hydrophila*, as natural inhabitants of the ecosystems, has acquired antibiotic resistance against a variety of beta-lactams (3). More seriously, colistin, the last resort against multi-drug resistant (MDR) microorganisms, was reported

**Peer Reviewers** Rui Bao, Sichuan University, Chengdu, China; Sattar Taheri-Araghi, California State University Northridge, Northridge, California, USA

Address correspondence to Zhiliang Sun, sunzhiliang1965@aliyun.com.

Leisheng Sun and Zonghan Jiang contributed equally to this article. Author order was determined both alphabetically and in order of increasing seniority.

The authors declare no conflict of interest.

See the funding table on p. 8.

10.1128/spectrum.00590-25   **1**

to show low potency to kill some *A. hydrophila* strains isolated from clinical samples or developed by lab culture (4). Therefore, the discovery of new agents, which are potent to sterilize colistin-resistant *A. hydrophila*, is necessary.

Membrane-permeabilizing antimicrobial peptides (AMPs) are known as a promising and effective strategy to treat bacterial infections, especially caused by drug-resistant strains (5, 6). Although thousands of AMPs have been discovered with high potency and broad-spectrum antimicrobial activity against bacteria, the toxicity and instability *in vivo* impede more investigations and further clinical trials (7, 8). Previously, a broad-spectrum, host cell-compatible antimicrobial peptide called D-CONGA (<u>D</u>-amino acid <u>con</u>sensus with <u>g</u>lycine <u>a</u>bsent; amino acid sequence: rrwarrlafafrr-amide) was screened and identified by synthetic molecular evolution from a peptide library containing 28,000 members (9). Subsequently, D-CONGA was modified with a polar glutamine inserted between the cationic and hydrophobic segments. This new variant, called D-Q7, showed a broader antimicrobial spectrum, improved antibiotic and antibiofilm activity with even lower residual toxicity compared to its parent peptide, D-CONGA (10). Therefore, it is encouraging to investigate whether D-Q7 is a potential candidate AMP against colistin-resistant *A. hydrophila*.

To study the colistin-resistant *A. hydrophila* strain, we have previously developed *A. hydrophila* strain called *23-C-23* with high-level colistin resistance (colistin MIC, 256 µg/mL) by serial passage of clinical colistin-susceptible *A. hydrophila* strain *WCX23* (colistin MIC, 1 µg/mL), which was isolated from farm-raised *Deinagkistrodon acutus* snakes during a fatal diarrhea outbreak in China (11, 12). *23-C-23* was also a multi-drug-resistant *A. hydrophila* strain due to significantly high MICs of many conventional antibiotics against it, while the antibacterial activity of D-Q7 (MIC = 4 µg/mL) against *23-C-23* is much more potent than colistin and other conventional antibiotics (Table 1). When *23-C-23* was treated with sub-MIC of D-Q7 for 4 and 8 hours and subjected to transmission electron microscope with high resolution, cell shrinkage and cell wall damage with organelles leakage of *23-C-23* were observed (Fig. 1A). Subsequently, we determined if reactive oxygen species (ROS) production is involved in D-Q7-induced bacteria death, given that ROS can damage a variety of macromolecules, including nucleic acids, proteins, and lipids, and compromise cell viability (13). Our results revealed that both *WCX23* and *23-C-23* produced significantly higher levels of ROS with D-Q7 treatment (Fig. 1B). Besides, we found that ATP production by both strains was obviously reduced with this peptide treatment, consistent with overall decreased bacteria viability (Fig. 1C). To further verify the mechanism of D-Q7 disrupting bacterial cell membranes, we investigated the effects of various components including phosphatidylglycerol (PG), phosphatidylcholine (PC), and lipopolysaccharide (LPS) in bacterial outer membranes on the antimicrobial efficacy of D-Q7 by using the exogenous addition method (14). The results indicated that the activity of D-Q7 against *23-C-23* gradually decreased with the increased PG or LPS concentrations. Specifically, the MIC of D-Q7 increased 4-fold and 16-fold when the concentration of exogenous LPS and PG reached 200 µg/mL, respectively. By contrast, the addition of exogenous PC decreased the MIC value of D-Q7. These

**TABLE 1** Minimum inhibitory concentration (MIC) of D-Q7 and antibiotics against different *Aeromonas hydrophila* strains

| Strain | MIC (µg/mL) of D-Q7 | | | | | | | |
|---|---|---|---|---|---|---|---|---|
| *23-C-23* | 4 | | | | | | | |
| *23-C-23:Δgene3832* | 1 | | | | | | | |
| *23-C-23:CΔgene3832* | 4 | | | | | | | |
| *23-C-23:ΔenvZ/ompR* | 0.5 | | | | | | | |
| *EGCT42* | 4 | | | | | | | |
| *RTF13* | 8 | | | | | | | |
| *WWO29* | 8 | | | | | | | |
| *23-C-23* | MIC (µg/mL) of antibiotic: | | | | | | | |
| | Penicillin | Lincomycin | Streptomycin | Kanamycin | Gentamicin | Tetracycline | Erythromycin | Azithromycin | Cefradine |
| | 4,096 | 512 | 4,096 | 2,048 | 16 | 8 | 16 | 32 | 512 |

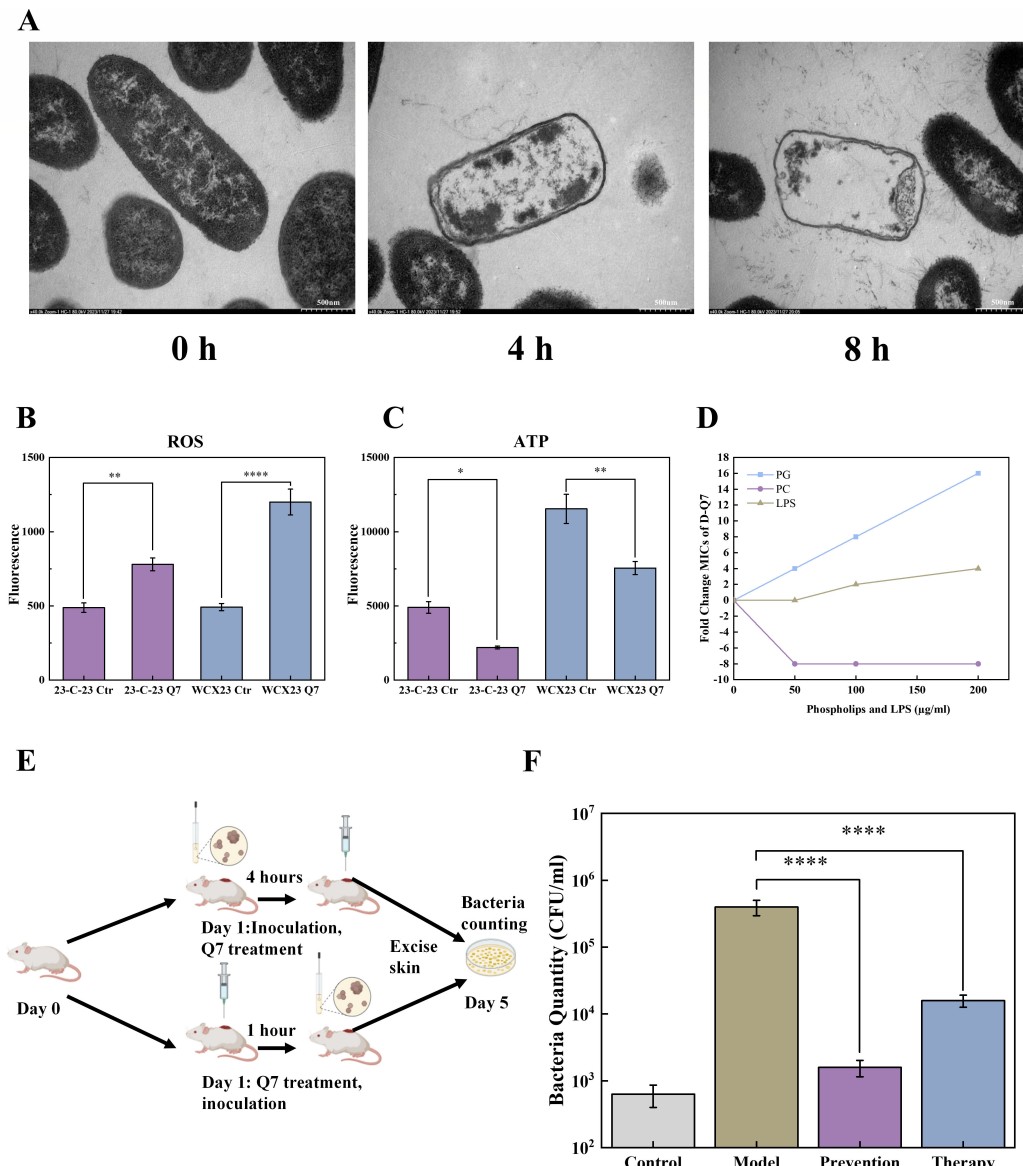

**FIG 1** D-Q7 has significant sterilizing activity against colistin-resistant *A. hydrophila 23-C-23*. (A) Colistin-resistant *A. hydrophila 23-C-23* was treated with 1 µg/mL D-Q7 for 4 and 8 hours, then the bacteria were observed by transmission electron microscope (TEM). *WCX23* and *23-C-23* were treated with 2 µg/mL D-Q7 or vehicle control (Ctr), then reactive oxygen species (ROS) and bacteria ATP were measured (B) and (C). (D) Effects of exogenous addition of phosphatidylglycerol (PG), phosphatidylcholine (PC), and lipopolysaccharide (LPS) (0–200 µg/mL) on the anti-23-C-23 activity of D-Q7, respectively. (E) A murine epicutaneous infection model was established. 23-C-23 inoculation after D-Q7 treatment was set as the prevention group; D-Q7 treatment after 23-C-23 inoculation was set as the therapeutic group. (F) On day 5, the skin infected on mice was extracted, determining the bacterial load. For (B, C, F) (n=3), values are expressed as mean ± SD. Compared with blank, *$P < 0.05$, **$P < 0.01$, ***$P < 0.001$, ****$P < 0.0001$.

results suggested that positively charged D-Q7 could selectively interact with negatively charged PG and LPS on the bacterial outer membrane, leading to rupture of the cell membrane and ultimately bacterial cell death, and that the addition of exogenous PC might mediate micellar interactions to improve peptide molecule accessibility, which enhances D-Q7 activity against *23-C-23*. Collectively, these findings indicated that D-Q7 could potently sterilize colistin-resistant *A. hydrophila 23-C-23 in vitro*.

Considering the promising activity of D-Q7 against *23-C-23 in vitro*, we sought to determine its function as an effective antibiotic *in vivo*. We opted for topical application of D-Q7 and established a murine epicutaneous infection model in Kunming mice infected intradermally with *23-C-23*. The bacteria burden was determined by

homogenizing the skin lesion with infection and plating samples on tryptone soy agar (TSA) plates on day 5. Topical application of 4 mg/mL D-Q7 on skin lesion after mice infection was set as the prophylactic group. While in the therapeutic group, mice were first infected with bacteria, and topical application of 4 mg/mL D-Q7 was conducted after 4 hours (Fig. 1D). The results showed that compared with vehicles (no treatment after infection), both prophylactic and therapeutic groups had significantly lowered the bacteria burden, with the prophylactic group presenting the lowest burden (Fig. 1E). Overall, these results indicated that low-dose D-Q7 had a long-lasting anti-*23-C-23* effect in an animal wound model.

Subsequently, to elucidate the mode of action of D-Q7 against *23-C-23*, we treated both *WCX23* and *23-C-23* strains with sub-MIC of D-Q7 for 4 and 8 hours, respectively, and bacterial RNA was extracted for sequencing. Transcriptomics analysis revealed that peptide incubation with either strain caused apparent alteration of gene expression as evidenced by the differentially expressed genes (DEGs) between the blank control and peptide-treated groups. Collectively, Kyoto Encyclopedia of Genes and Genomes (KEGG) pathway analysis showed that these genes are enriched in a variety of metabolic pathways and are particularly enriched in ABC transporter proteins, which are a series of transmembrane transporter systems designed to transport a wide range of molecules either across membranes into or export a wide range of molecules, including antibiotics, sugars, and lipids (15) (Fig. 2A and B). Given that the disruption of bacterial membranes is the most conventional mechanism underlying bacterial killing by AMPs such as colistin (16), we focused on transmembrane proteins. Specifically, we found *gene3832*, whose expression was reported to modulate membrane permeability as porin protein, was significantly upregulated after treatment with D-Q7 (Fig. 2C). Consistently, quantitative real-time PCR (qRT-PCR) analysis revealed that exposure to D-Q7 induced significant upregulation of gene3832 expression across four distinct *A. hydrophila* isolates (*23-C-23*, *EGCT42*, *RTF13*, and *WWO29*) (Fig. 2F). This conserved transcriptional response among genetically diverse strains strongly suggested that *gene3832* may encode a previously uncharacterized stress-response element associated with AMP exposure. Thus, we deduced that the upregulation of *gene3832* could decrease membrane permeability, a potential defense mode in response to external stimuli such as D-Q7 for bacterial survival (17).

To validate the function of *gene3832* in defending the attack from D-Q7, we knocked out *gene3832* in *23-C-23* (*23-C-23:Δ3832*) and also complemented *gene3832* back in the knockout strain (*23-C-23:CΔ3832*). We subjected these different *23-C-23* strains to D-Q7 and found that the MIC for *23-C-23:Δ3832* was 1 µg/mL, lower than wild-type *23-C-23* and *23-C-23:C3832* (4 µg/mL) (Table 1). Correspondingly, strain 23-C-23:Δ3832 demonstrated markedly increased membrane permeability relative to the wild-type 23-C-23 and 23-C-23:CΔ3832 (Fig. 2D). *EnvZ/OmpR* can modulate the outer-membrane porins such as OmpF and OmpC in response to adverse stimuli for self-protection, and it was recently reported to regulate the expression of *gene3832* to decrease outer membrane permeability in response to colistin (15, 18). Therefore, we also tested D-Q7 activity against the *EnvZ/OmpR* knockout *23-C-23* strain and found the MIC was only 0.5 µg/mL, much lower than wild-type *23-C-23* (Table 1). These results were consistent with our deduction that the mode of antibacterial effect of D-Q7 was membrane disruption, in which the expression of gene3832 as an autotransporter might decrease membrane permeability as a potential defense mode. In addition, to further investigate the function of *gene3832* in *23-C-23*, we compared biofilm formation ability between these three strains (*23-C-23*; *23-C-23:Δ3832*; *23-C-23:CΔ3832*). The results showed that *gene3832* knock-out *23-C-23* had decreased biofilm ability compared with other strains after D-Q7 treatment (Fig. 2E). Collectively, these results indicated that *gene3832* participated in *23-C-23* survival and growth, suggesting *gene3832* as a potential target for the development of new agents to combat MDR *A. hydrophila* infection in the future.

In summary, we present our findings that D-Q7 is a potent antibacterial agent that kills colistin-resistant *A. hydrophila 23-C-23 in vitro*, and it has extraordinarily antibacterial

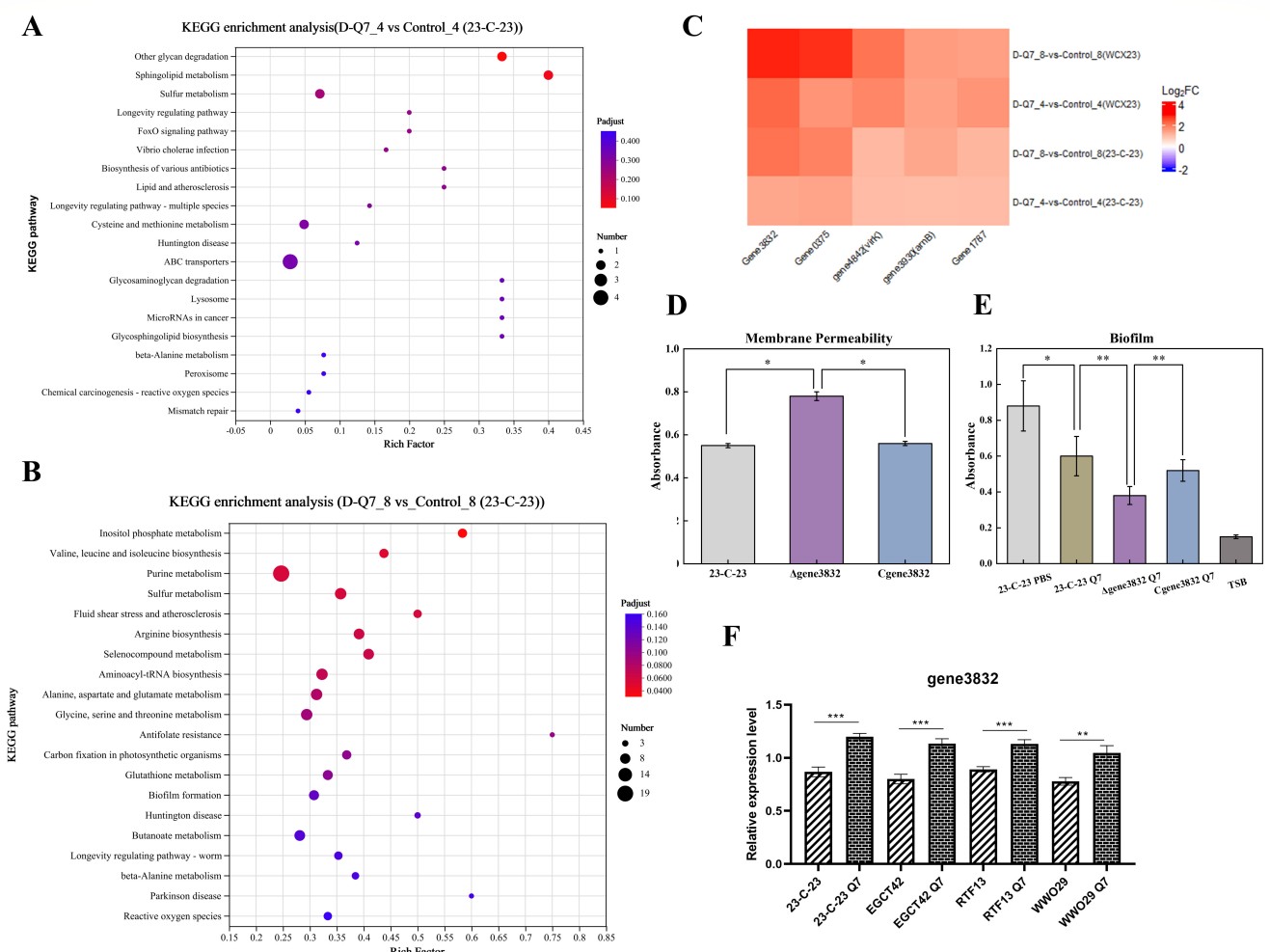

**FIG 2** *Gene3832* is involved in D-Q7 activity against *23-C-23* strain. *23-C-23* strains were treated with 1 µg/mL D-Q7 for 4 and 8 hours, and the RNA was extracted for sequencing. KEGG analyses were shown for 4-hour (A) and 8-hour (B) treatment, respectively. (C) Heat map was shown for the transcriptional level change of *23-C-23* and *WXC23* strains for specific genes with D-Q7 treatment for 4 or 8 hours. (D) Strain 23-C-23:Δ3832, wild-type 23-C-23, and 23-C-23:CΔ3832 exhibit marked alterations in membrane permeability. (E) Strain 23-C-23:Δ3832, wild-type 23-C-23 and 23-C-23:CΔ3832 undergo altered biofilm formation capacity after D-Q7 treatment. (F) Relative transcriptional levels of gene3832 in different *Aeromonas hydrophila* strains treated with D-Q7 as determined by quantitative RT-PCR. For (D, E, F) (n=3), values are expressed as mean ± SD. Compared with blank, *$P < 0.05$, **$P < 0.01$, ***$P < 0.001$, ****$P < 0.0001$.

efficacy in the mouse epicutaneous model of *23-C-23* infection. Mechanistic studies revealed that D-Q7 interacts with PG and LPS on the bacterial cell membrane and induces an increase in intracellular ROS levels and a decrease in ATP, leading to cell membrane rupture and bacterial death. RNA sequencing of D-Q7-treated *23-C-23* identified several genes as potential targets. Specifically, *gene3832*, a transmembrane autotransporter as predicted, is associated with bacteria resistance to this peptide. The knockout of *gene3832* endows *23-C-23* with greater susceptibility to D-Q7, while *gene3832* complementation restores the antimicrobial effect of this peptide against *23-C-23*. Importantly, we found the presence of *gene3832* in *23-C-23* maintained biofilm formation and membrane permeability of this colistin-resistant *A. hydrophila* strain. Our study first provided a promising peptide lead compound for the development of colistin-resistant *A. hydrophila* drugs and demonstrated that *gene3832* in *A. hydrophila* may act as a potential resistance regulator to modulate bacterial resistance to drugs.

## Strains, plasmids, and peptide

The *A. hydrophila* strain *WCX23* was isolated from a snake with fatal diarrhea in the city of Wangcheng, Hunan province in 2017 and was identified as ST516 via multilocus sequence typing; the MDR *A. hydrophila* strain *23-C-23* was selected in the susceptible *A. hydrophila WCX23* strain by serial daily passages on TSA with increasing concentrations of colistin sulfate; the *A. hydrophila* strains *EGCT42*, *RTF13*, and *WWO29* used in this study were isolated from environmental water sample, sick bullfrog, and sick seabass, respectively, collected by our laboratory (Hunan Engineering Research Center of Veterinary Drug, Changsha, Hunan, China). The mutant strains (*23-C-23:ΔEnvZ/OmpR* and *23-C-23:Δgene3832*) and the complemented strain (*23-C-23:CΔgene3832*) were constructed in our previously published work. In brief, *23-C-23:ΔEnvZ/OmpR* and *23-C-23:Δgene3832* were generated by deleting the EnvZ/OmpR operon and gene3832, respectively, from the parental strain *23-C-23*, whereas *23-C-23:CΔgene3832* is the *23-C-23:Δgene3832* knockout complemented with a functional copy of *gene3832*. The construction methods and relevant characteristics of these strains are described in detail in our earlier publication (12, 15). All strains were grown on TSA (Oxoid, Basingstoke, UK) or cultured in tryptone soy broth (TSB; Oxoid, Basingstoke, UK at 28°C. Peptide D-Q7 was purchased from GL Biochem Ltd. (Shanghai). All the antibiotics used in this study were purchased from MCE Ltd (Monmouth Junction, NJ, USA).

## Determination of minimal inhibitory concentration (MIC)

The minimal inhibitory concentration of D-type polypeptide (D-Q7) against *A. hydrophila* was determined using the micro broth dilution method recommended by the Clinical and Laboratory Standards Institute (CLSI), USA. The bacterial concentration was adjusted to $10^5$ CFU/mL using a turbidimeter. Then, 100 µL of drug was added to wells 1–10 of the 96-well plate, while 100 µL of Muller-Hinton Broth (MHB) was added to wells 11–12. Subsequently, 100 µL of bacterial suspension was added to wells 1–12. The plates were then incubated at 28°C for 16–24 hours, and the results were observed. The MIC results were interpreted based on the CLSI MD100-ED30 (2024) guidelines. Three biological replicates were performed.

## Transmission electron microscopy

The concentration of *A. hydrophila* was adjusted to $10^5$ CFU/mL using a turbidimeter. The bacteria were treated with 1 mg/L D-Q7 for 4 hours, then resuspended, washed twice with phosphate-buffered saline (PBS), and centrifuged at 8,000 rpm for 10 minutes. The supernatant was discarded, and the bacterial pellet was fixed overnight at 4°C in tissue fixative and analyzed by transmission electron microscopy (Hitachi, HT7800).

## Reactive oxygen species (ROS) measurement

The ROS level in bacteria was measured using a ROS detection kit(S0033M) (Beyotime Biotechnology, Shanghai, China). *A. hydrophila* was cultured until OD600 nm = 0.5, then 0.25× MIC, 0.5× MIC, and 1× MIC D-Q7 were added, mixed, and incubated for 1 hour. After incubation, the 2′,7′-dichlorofluorescein diacetate probe was loaded for 30 minutes with shaking every 3 minutes. After loading, the probe was washed three times with 1× PBS, and the ROS levels were measured using excitation at 488 nm and emission at 525 nm. Three biological replicates were performed for each group.

## Effect of bacterial cell membrane components on the anti-*23-C-23* activity of D-Q7

The checkerboard broth dilution method was used to study the effect of bacterial cell membrane components on the antimicrobial efficacy of D-Q7. D-Q7 was prepared as a stock solution of 1,024 µg/mL in sterile water. The lipids (PC and PG) were purchased from AVT (Shanghai) Pharmaceutical Technology Co., Ltd. First, different concentrations

of D-Q7, PC, LPS, and PG were added into 96-well plates, and then 100 µL of *23-C-23* bacterial suspension ($1 \times 10^5$ CFU/mL) was added, and the results were read after incubation at 28°C for 16–24 hours. MHB was used as a negative control, and *23-C-23* bacterial suspension was used as a positive control.

## ATP measurement

The bacterial ATP level was measured using an enhanced ATP assay kit (S0027) (Beyotime Biotechnology, Shanghai, China). *A. hydrophila* was cultured until OD600 nm = 0.5, then 0.25× MIC, 0.5× MIC, and 1× MIC D-Q7 were added, mixed, and incubated at 28°C for 1 hour. After incubation, the bacteria were washed three times with 1× PBS, centrifuged, and resuspended. The bacterial suspension was treated with 200 µL of 15 mg/mL lysozyme, vortexed, and the supernatant was collected. The sample and detection reagent were mixed in a 1:1 ratio in a black 96-well plate and measured using a luminometer. Three biological replicates were set for each group.

## Transcriptome analysis

Briefly, the strains were cultured to log-phase (OD600 nm = 0.6 ± 0.05) at 28°C, and then the cells were harvested and washed twice by sterilized PBS. Total RNA was extracted using Qiagen RNeasy Mini kits (Qiagen, Hilden, Germany) and quantified using a NanoDrop 2000 spectrophotometer (Thermo Fisher, Waltham, MA, USA). rRNA was depleted, and cDNA libraries were prepared. The cDNA libraries were further sequenced by the Illumina Hiseq 2000 system (Majorbio, Shanghai, China). DEGs were identified by gene expression-level analysis using the Fragments Per Kilobase of transcript per Million mapped reads method with $P$-values ≤ 0.05 and fold change (FC) values ≥ 2 ($\log2_2$ FC ≥ 1 or $\log2_2$ FC ≤ −1). Gene Ontology and KEGG enrichment pathway analysis of DEGs was performed using the R package (https://bioconductor.org/packages/edgeR).

## Animals and treatments

Experimental animals were obtained from Hunan SJA Laboratory Animal Co., Ltd. Kunming mice were acclimated for one week in a controlled environment with a 12-hour light/dark cycle, a relative humidity of 60% ± 10%, and a temperature of 24 ± 2°C. All mice were anesthetized using tribromoethanol. The mice (6–8 weeks old) were weighed and randomly assigned to four groups, each consisting of eight mice: a blank group, a model group, a prophylactic group, and a therapeutic group. The mice were anesthetized, and a 5 mm diameter wound was created on the mid-back using ophthalmic forceps and iris scissors. For the blank group, we did not infect *23-C-23* on the wound or apply any drugs. For the model group, we infected the wound with 10 µL suspension of *23-C-23* (at a concentration of $10^8$ CFU/mL) without any drug treatment. For the prophylactic group of mice, we applied 10 µl of 4 mg/mL D-Q7 to circular wounds in advance at the wound site, and after 1 hour, we applied a 10 µL suspension of *23-C-23* (at a concentration of $10^8$ CFU/mL) to infect the circular wounds of mice. For the therapeutic group, we infected the wound with 10 µL suspension of *23-C-23* (at a concentration of $10^8$ CFU/mL), and after 4 hours, 10 µL 4 mg/mL D-Q7 was added onto the circular wound.

## Measurement of bacterial counts at the wound site

On the 5th day of treatment, four mice were randomly selected from each group and euthanized. A square piece of skin tissue $1 \times 1$ cm$^2$, centered on the wound, was excised using surgical scissors and placed in a 2 mL grinding tube containing 1 mL of physiological saline and grinding beads. The tissue was subjected to 60 Hz shaking and grinding for 180 seconds, followed by the collection of the supernatant. After gradient dilution, the supernatant was evenly plated on TSA medium containing 8 µg/mL colistin. The tissue was incubated at 28°C for 16–24 hours, and the bacterial count of *23-C-23* in the wound tissue was determined.

## Biofilm formation assay

The strains were cultured to log-phase (OD600 nm = 0.6 ± 0.05) at 28℃, and adjusted to an OD 600 nm of 0.1, and 200 µL of culture (1:100 dilution) was dispensed into 96-well plates and incubated at 28℃. A 200 µL aliquot of fresh TSB was added to the plates as a blank control. After 24-hour incubation, the medium was discarded, and the plates were washed three times with sterile PBS. The biofilm was fixed with 200 µL of methanol for 15 minutes and incubated with 200 µL of 1% crystal violet for 15 minutes. After several washes with ddH$_2$O and air-dried, the dried biofilm was solubilized in 200 µL 95% ethanol for 10 minutes. The absorbance was measured at 595 nm.

## Determination of bacterial outer membrane permeability

N-Phenyl-1-naphthylamine (NPN) was used to assess the integrity of the outer membranes of *23-C-23*. The bacterial suspension was diluted 1:100 in LB broth and incubated at 28℃ until reaching an OD600nm of 0.6. The cells were then centrifuged, resuspended, and the OD600nm was adjusted to 0.5 using PBS. NPN was added at a final concentration of 10 µM, and the mixture was incubated for 30 minutes at 28℃ in the dark. Added final concentration of 0.25× MIC, 0.5× MIC, and 1× MIC D-Q7 incubated for 1 hour. Fluorescence intensity was measured using a fluorescence plate reader with excitation wavelengths of 350 nm and emission wavelengths of 420 nm (NPN). Three biological replicates were prepared. NPN (P110559) was purchased from Aladdin Biochemical Technology, Shanghai, China.

## Statistical analyses

All the experiments were tested at least three times. All data were expressed as the mean ± SD. Student's *t*-test and analysis of variance were used to process and analyze routine data. *P* < 0.05 represents statistically significant.

## ACKNOWLEDGMENTS

This work was supported by the Hunan Natural Science Foundation (2023JJ30301) and the Hunan Provincial Key Laboratory of Anti-Resistance Microbial Drugs, the Third Hospital of Changsha (No: 2023TP1013).

## AUTHOR AFFILIATIONS

[1]Key Laboratory of Study and Discovery of Small Targeted Molecules of Hunan Province, Department of Pharmacy, School of Medicine, Hunan Normal University, Changsha, PR China
[2]Hunan Provincial Key Laboratory of Anti-Resistance Microbial Drugs, The Third Hospital of Changsha, Changsha, PR China
[3]Hunan Engineering Research Center of Livestock and Poultry Health Care, College of Veterinary Medicine, Hunan Agricultural University, Changsha, PR China
[4]Hunan Institute of Animal and Veterinary Science, Hunan Academy of Agricultural Sciences, Changsha, PR China

## AUTHOR ORCIDs

Leisheng Sun http://orcid.org/0000-0003-0358-1421
Zhiliang Sun http://orcid.org/0000-0002-6612-6541

## FUNDING

| Funder | Grant(s) | Author(s) |
| --- | --- | --- |
| The Hunan Provincial Key Laboratory of Anti-ResistanceMicrobial Drugs, the Third Hospital of Changsha | 2023TP1013 | Leisheng Sun |

| Funder | Grant(s) | Author(s) |
|---|---|---|
| Natural Science Foundation of Hunan Province | 2023JJ30301 | Zhiliang Sun |

## AUTHOR CONTRIBUTIONS

Leisheng Sun, Conceptualization, Data curation, Formal analysis, Funding acquisition, Investigation, Methodology, Resources, Supervision, Writing – original draft | Zonghan Jiang, Data curation, Investigation, Methodology, Software, Validation | Dingding Li, Data curation, Formal analysis, Methodology, Writing – original draft | Yao Tian, Data curation, Formal analysis, Investigation | Junqi Liu, Resources, Software, Visualization, Writing – original draft | Zhiliang Sun, Funding acquisition, Investigation, Project administration, Resources, Supervision, Writing – review and editing

## ETHICS APPROVAL

All experimental procedures and sample collections were conducted in accordance with the Chinese Animal Welfare Guidelines and approved by the Institutional Animal Care and Use Committee of Hunan Agricultural University.

## ADDITIONAL FILES

The following material is available online.

### Open Peer Review

**PEER REVIEW HISTORY (review-history.pdf).** An accounting of the reviewer comments and feedback.

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
