## [Reviewer comments · Microbiology Spectrum]

Microbiology Spectrum

Killing of colistin-resistant *Aeromonas hydrophila* by a synthetical peptide

Leisheng Sun, Zonghan Jiang, Dingding Li, Yao Tian, Junqi Liu, and Zhiliang Sun

Corresponding Author(s): Zhiliang Sun, Hunan Agricultural University

Review Timeline:

Submission Date:	February 26, 2025
Editorial Decision:	March 31, 2025
Revision Received:	May 29, 2025
Accepted:	June 13, 2025

Editor: Jinshui Lin

Reviewer(s): Disclosure of reviewer identity is with reference to reviewer comments included in decision letter(s). The following individuals involved in review of your submission have agreed to reveal their identity: Rui Bao (Reviewer #1); Sattar Taheri-Araghi (Reviewer #2)

Transaction Report:

DOI: <https://doi.org/10.1128/spectrum.00590-25>

Re: Spectrum00590-25 (**Killing of colistin-resistant *Aeromonas hydrophila* by a synthetic peptide**)

Dear Prof. Zhiliang Sun:

Thank you for the privilege of reviewing your work. Below you will find my comments, instructions from the Spectrum editorial office, and the reviewer comments.

Revision Guidelines

Sincerely,
Jinshui Lin
Editor
Microbiology Spectrum

Reviewer #1 (Comments for the Author):

Major :

1. The study primarily focuses on a single laboratory-induced strain (23-C-23). To enhance clinical relevance, the authors should expand their investigation to include a broader panel of clinical isolates, including strains with varying levels of colistin resistance. Additionally, exploring the synergistic effects of D-Q7 with other antibiotics would strengthen the translational potential of this study.

2. While the study identifies gene3832 as a key resistance factor, the specific molecular mechanisms by which it confers resistance and interacts with D-Q7 remain unclear. Further investigation into the specific function of gene3832 such as protein-level validation and functional assays demonstrating gene3832's role in membrane permeability are needed.
3. Although the manuscript shows that D-Q7 disrupts bacterial membranes and induces ROS, the detailed mechanism of how D-Q7 interacts with the bacterial membrane and causes cell death is not fully elucidated. Further studies to understand the specific killing mechanisms are necessary.
4. While transcriptomic analysis identified gene3832, further validation through techniques like qRT-PCR or proteomics is required to confirm the differential gene expression and to provide a more robust mechanistic framework.

Minor:

1. The resolution of figures, particularly TEM images, is insufficient. Scale bars should be included, and higher-resolution versions are required. The color schemes in bar charts should be improved for visual clarity. All figures and tables should be reviewed to ensure consistent font styles and increased image resolution.
2. The formatting throughout the text needs to be standardized. Ensure spaces between numbers and units (e.g., OD600 nm). Correct the inconsistency in MIC units between the text (mg/L) and Table 1 ($\mu\text{g/ml}$).
3. Provide detailed descriptions of all experimental procedures, including bacterial strains, antimicrobial susceptibility testing methods, animal infection models, and transcriptomic analysis (DEG screening thresholds). Clarify the treatment method for the blank control group. Include positive controls (e.g., other antimicrobial peptides or antibiotics) in relevant experiments to directly compare efficacy.

Reviewer #2 (Comments for the Author):

This manuscript investigates the antimicrobial activity of a synthetic D-type peptide, D-Q7, against a colistin-resistant strain of *Aeromonas hydrophila* (23-C-23). The authors demonstrate that D-Q7 is highly effective both in vitro and in a murine epicutaneous infection model, significantly reducing bacterial burden. They further identify a transmembrane autotransporter gene, gene3832, as a potential modulator of peptide resistance, supported by transcriptomic analysis and functional validation via gene knockout and complementation. The findings offer a potential new strategy for combating drug-resistant *A. hydrophila*, with implications for peptide-based therapeutics.

I commend the authors for their comprehensive and well-structured experimental design, which approaches the central question from multiple angles. The work includes antimicrobial susceptibility testing, high-resolution microscopy, ROS and ATP assays, in vivo efficacy evaluation, transcriptomic profiling, and genetic validation. This breadth of methodology provides strong technical support for the conclusions. The condensed and focused writing style is effective for an Observation-type article, and the key message is communicated clearly.

Major Concerns:

Justification of Strain and Peptide Selection: While D-Q7 has been previously characterized and is a rational choice, the manuscript would benefit from a clearer explanation of why this particular peptide was selected to test against *A. hydrophila*, beyond citing past work. More importantly, the authors should discuss the broader antibiotic susceptibility profile of the colistin-resistant 23-C-23 strain. Since colistin resistance does not necessarily imply pan-resistance, it would help contextualize the clinical significance of D-Q7's activity - is D-Q7 filling an unmet need, or would standard antibiotics still work?

Mechanistic Interpretation of gene3832: While gene3832 is convincingly shown to affect D-Q7 susceptibility, the claim that it represents a mechanism of resistance could be toned down or clarified. The study relies on transcriptomic association and knockout data, but additional mechanistic insight (e.g., functional characterization of the protein product) is missing. Suggesting it as a potential resistance modulator is appropriate, but stronger language should be reserved unless more direct mechanistic evidence is available.

Comparison to Other Antibiotics: The study compares D-Q7 to colistin, but not to other antibiotics (e.g., aminoglycosides, quinolones). Including a brief comparison or referencing prior susceptibility profiles would help readers understand how D-Q7 fits into the current therapeutic landscape.

Minor Suggestions:

The writing is generally clear, but would benefit from language polishing in several areas to improve flow and precision. For example, phrasing like "the bacteria burden was determined" could be reworded for clarity.

Figure captions could be more descriptive and self-contained, especially for readers skimming the figures.

A short, one-sentence rationale for selecting D-Q7 in the Introduction would help orient readers unfamiliar with the peptide.

Dear Reviewer 1,

Thank you very much for your valuable comments. Your Suggestions have helped us to improve the deficiencies in the manuscript and increase the rigor and readability of the article. We have completed the revision and have responded to your comments point by point as follows:

Major:

1. The study primarily focuses on a single laboratory-induced strain (23-C-23). To enhance clinical relevance, the authors should expand their investigation to include a broader panel of clinical isolates, including strains with varying levels of colistin resistance. Additionally, exploring the synergistic effects of D-Q7 with other antibiotics would strengthen the translational potential of this study.

We sincerely appreciate your constructive suggestions for enhancing the clinical relevance of our study. In response, we have made the following key revisions:

1. Expansion of the isolate panel. Three additional *Aeromonas hydrophila* isolates—*EGCT42* (from environmental water sample), *RTF13* (from bullfrog), and *WWO29* (from seabass)—have been incorporated into the study. Minimum inhibitory concentrations (MICs) of D-Q7 against all isolates are now reported in Table 1, confirming the peptide's broad-spectrum activity across diverse backgrounds.

MIC (µg/ml)	D-Q7
23-C-23	4
EGCT42	4
RTF13	8
WWO29	8

2. Synergy studies. We fully agree with the reviewer's recommendation to explore the combinatorial activity of D-Q7 with conventional antibiotics. For this study, we mainly focus on the activity of D-Q7 alone, but we will initiate a series of D-Q7 combination assays with multiple antibiotics, and the complete results will be presented in a subsequent, dedicated study.

2. While the study identifies gene3832 as a key resistance factor, the specific molecular mechanisms by which it confers resistance and interacts with D-Q7 remain unclear. Further investigation into the specific function of gene3832 such as protein-level validation and functional assays demonstrating gene3832's role in membrane permeability are needed.

Thank you for your valuable suggestions! You pointed out the need to further investigate the specific function of *gene3832*, such as protein level validation and functional assay is crucial to prove the role of *gene3832* in membrane permeability. In this study, we mainly examined the **membrane permeability level** and biofilm formation of 23-C-23:Δ3832 as well as 23-C-23:CΔ3832, but did not perform protein level validation of gene3832. This is due to the fact that the expression, isolation and purification of novel bacterial protein gene3832 take a long time, and the antibody against gene3832 is inaccessible on the market, which is always an obstacle for bacterial protein validation. In addition, the experimental design in this study was more focused on resolving the functional effects of gene3832 on cell membrane permeability by genetical method (knockout/complementation). We fully agree with your suggestion and would like to continue subsequent studies in our future publication for the investigation of *gene3832* at the protein level to more comprehensively elucidate the molecular mechanism of gene3832's interaction with D-Q7.

3. Although the manuscript shows that D-Q7 disrupts bacterial membranes and induces ROS, the detailed mechanism of how D-Q7 interacts with the bacterial membrane and causes cell death is not fully elucidated. Further studies to understand the specific killing mechanisms are necessary.

We appreciate your suggestion to clarify the molecular events that link D-Q7-membrane interaction to bacterial killing. Previous work indicated that some antimicrobial peptide could interacted with phospholipid components in bacterial outer membrane and disrupt membrane integrity. To this end, we explored the effect

of exogenously added cell membrane components including phosphatidylglycerol (PG), phosphatidylcholine (PC), and lipopolysaccharide (LPS) present in bacterial outer membranes on D-Q7 anti-23-C-23 activity; the corresponding data are presented in Figure 1E. A summary of the new findings is provided below.

1. Experimental design

To determine whether D-Q7 targets specific membrane lipids, we measured its MIC against the colistin-resistant strain 23-C-23 in the presence of increasing concentrations (0-200 $\mu\text{g/ml}$) of three purified membrane components: phosphatidyl-glycerol (PG), lipopolysaccharide (LPS) and phosphatidyl-choline (PC).

2. Key observations

When PG or LPS was added externally, the MIC of D-Q7 rose in a concentration-dependent manner. Most strikingly, PG at 200 $\mu\text{g/ml}$ increased the MIC from 4 to 64 $\mu\text{g/ml}$ (16-fold). In contrast, supplementation with PC reduced the MIC.

These results demonstrate that cationic D-Q7 preferentially interacts with the negatively charged lipids PG and LPS, whereas the zwitterionic PC does not compete but may facilitate peptide accessibility by micelle formation. The data support a model in which electrostatic binding of D-Q7 to PG/LPS is a critical first step that promotes membrane insertion, rupture and subsequent cell death. Excess exogenous PG/LPS “titrates” the peptide and attenuates killing, whereas PC blunts such titration and increases the freely available peptide fraction. We believe these new experiments substantially advance our understanding of how D-Q7 interacts with bacterial membranes and address your concern. Thank you very much for the thoughtful feedback that helped us strengthen the manuscript.

4. While transcriptomic analysis identified gene3832, further validation through techniques like qRT-PCR or proteomics is required to confirm the differential gene expression and to provide a more robust mechanistic framework.

Thank you for your valuable suggestions! We fully agree that independent validation of *gene 3832* expression is essential. Accordingly, we performed quantitative RT-PCR

(qRT-PCR) on four distinct *Aeromonas hydrophila* isolates to confirm the transcriptomic findings. The results showed that the expression of *gene3832* was significantly up-regulated in all the four strains after D-Q7 treatment. The corresponding data are presented in Figure 2F. We believe the new qRT-PCR data directly address the concern and enhance the robustness of the manuscript. Thank you for asking us to strengthen this aspect of the study.

Minor:

1. The resolution of figures, particularly TEM images, is insufficient. Scale bars should be included, and higher-resolution versions are required. The color schemes in bar charts should be improved for visual clarity. All figures and tables should be reviewed to ensure consistent font styles and increased image resolution.

We appreciate your careful assessment of our figures and have comprehensively revised the manuscript to meet publication quality standards. These revisions address all concerns about image clarity, scale bars, color readability, and font consistency. We thank the reviewer for helping us improve the visual quality of the manuscript.

2. The formatting throughout the text needs to be standardized. Ensure spaces between numbers and units (e.g., OD600 nm). Correct the inconsistency in MIC units between the text (mg/L) and Table 1 (µg/ml).

Thank you for drawing our attention to these formatting details. In accordance with your recommendations, we have carefully standardised the formatting throughout the manuscript (we all use µg/ml instead of mg/L) and performed a page-by-page check to ensure that no inconsistencies remain. We sincerely appreciate your assistance in improving the professional presentation of our paper.

3. Provide detailed descriptions of all experimental procedures, including bacterial strains, antimicrobial susceptibility testing methods, animal infection models, and transcriptomic analysis (DEG screening thresholds). Clarify the treatment method for the blank control group. Include positive controls (e.g., other antimicrobial peptides or antibiotics) in relevant experiments to directly compare efficacy.

We thank you for requesting greater methodological clarity. In the revised manuscript we have expanded the Materials and Methods section. The details are as follows:

1. Strains, plasmids and peptide

The *Aeromonas hydrophila* strain *WCX23* was isolated from a snake with fatal diarrhea in the city of Wangcheng, Hunan province in 2017 and was identified as ST516 via multilocus sequence typing (MLST); the multi-drug resistant *Aeromonas hydrophila* strain *23-C-23* was selected in the susceptible *A. hydrophila WCX23* strain by serial daily passages on TSA with increasing concentrations of colistin sulfate; The *A. hydrophila* strains *EGCT42*, *RTF13*, and *WWO29* used in this study were isolated from environmental water sample, sick bullfrog and sick seabass respectively collected by our laboratory (Hunan Engineering Research Center of Veterinary Drug, Changsha, Hunan, China). The mutant strains (*23-C-23:ΔEnvZ/OmpR* and *23-C-23:Δgene3832*) and the complemented strain (*23-C-23:CΔgene3832*) were constructed in our previously published work. In brief, *23-C-23:ΔEnvZ/OmpR* and

23-C-23:*Δgene3832* were generated by deleting the EnvZ/OmpR operon and gene3832, respectively, from the parental strain 23-C-23, whereas 23-C-23:*CΔgene3832* is the 23-C-23:*Δgene3832* knockout complemented with a functional copy of *gene3832*. The construction methods and relevant characteristics of these strains are described in detail in our earlier publication[1]. All strains were grown on tryptone soy agar (TSA; Oxoid, Basingstoke, United Kingdom) or cultured in tryptone soy broth (TSB; Oxoid, Basingstoke, United Kingdom) at 28°C. D-Q7 was purchased from GL Biochem (Shanghai) Ltd. All the antibiotics were purchased from MCE, (Monmouth Junction, NJ) Ltd., USA.

2. Determination of Minimal Inhibitory Concentration (MIC)

The minimal inhibitory concentration of D-type polypeptide (D-Q7) against *A. hydrophila* was determined using the micro broth dilution method recommended by the Clinical and Laboratory Standards Institute (CLSI), USA. The bacterial concentration was adjusted to 10^5 CFU/mL using a turbidimeter. Then, 100μL of drug was added to wells 1-10 of the 96-well plate, while 100μL of Muller-Hinton Broth (MHB) was added to wells 11-12. Subsequently 100μL of bacterial suspension was added to wells 1-12. The plates were then incubated at 28°C for 16-24 hours, and the results were observed. The MIC results were interpreted based on the CLSI MD100-ED30 (2024) guidelines. Three biological replicates were performed

3. Animals and Treatments

Experimental animals were obtained from Hunan SJA Laboratory Animal Co., Ltd. Kunming mice were acclimated for one week in a controlled environment with a 12-hour light/dark cycle, a relative humidity of 60% ± 10%, and a temperature of 24 ± 2°C. All mice were anesthetized using tribromoethanol. The mice (6-8 weeks old) were weighed and randomly assigned to four groups, each consisting of eight mice: a blank group, a model group, a prophylactic group, and a therapeutic group. The mice were anesthetized, and a 5 mm diameter wound was created on the mid-back using ophthalmic forceps and iris scissors. For the blank group, we did not

infect 23-C-23 on the wound or applied any drugs. For the model group, we infected the wound with 10 μ L suspension of 23-C-23 (at a concentration of 10^8 CFU/ml) without any drug treatment. For the prophylactic group of mice, we applied 10 μ L of 4 mg/ml D-Q7 to circular wounds in advance at the wound site, and after 1 hour, we applied a 10 μ L suspension of 23-C-23 (at a concentration of 10^8 CFU/ml) to infect the circular wounds of mice. For the therapeutical group, we infected the wound with 10 μ L suspension of 23-C-23 (at a concentration of 10^8 CFU/ml), and after 4 hours, 10 μ L 4 mg/ml D-Q7 were added onto circular wound.

4. Measurement of Bacterial Counts at the Wound Site

On the 5th day of treatment, four mice were randomly selected from each group and euthanized. A square piece of skin tissue approximately 1×1 cm², centered on the wound, was excised using surgical scissors and placed in a 2 ml grinding tube containing 1 mL of physiological saline and grinding beads. The tissue was subjected to 60 Hz shaking and grinding for 180 seconds, followed by collection of the supernatant. After gradient dilution, the supernatant was evenly plated on TSA medium containing 8 μ g/mL colistin. The tissue was incubated at 28°C for 16-24 hours, and the bacterial count of 23-C-23 in the wound tissue was determined. All experimental procedures and sample collections were conducted in accordance with the Chinese Animal Welfare Guidelines and approved by the Institutional Animal Care and Use Committee of Hunan Agricultural University.

5. Transcriptome Analysis

Briefly, the strains were cultured to log-phase ($OD_{600\text{ nm}} = 0.6 \pm 0.05$) at 28°C, and then the cells were harvested and washed twice by sterilized PBS. Total RNA was extracted using Qiagen RNeasy Mini kits (Qiagen, Hilden, Germany), and quantified using a NanoDrop 2000 spectrophotometer (Thermo Fisher, Waltham, MA, United States). rRNA was depleted, and cDNA libraries were prepared. The cDNA libraries were further sequenced by Illumina HiSeq 2000 system (Majorbio, Shanghai, China). Differentially expressed genes were identified by gene expression-level analysis using

the FPKM (Fragments Per Kilobase of transcript per Million mapped reads) method with p-values ≤ 0.05 and fold change (FC) values ≥ 2 ($\log_2 \text{FC} \geq 1$ or $\log_2 \text{FC} \leq -1$). Gene Ontology (GO) and Kyoto Encyclopedia of Genes and Genomes (KEGG) enrichment pathway analysis of DEGs was performed using the R package (<https://bioconductor.org/packages/edgeR>).

We believe these additions will meet your request for detailed methodological transparency and head-to-head benchmarks, thereby strengthening the mechanistic and translational foundation of the study. We appreciate your guidance in improving the manuscript.

Reference

1. Liu, J., et al., *Various Novel Colistin Resistance Mechanisms Interact To Facilitate Adaptation of Aeromonas hydrophila to Complex Colistin Environments*. *Antimicrob Agents Chemother*, 2021. **65**(7): p. e0007121.

Dear Reviewer 2,

Thank you very much for your valuable comments. Your Suggestions have helped us to improve the deficiencies in the manuscript and increase the rigor and readability of the article. We have completed the revision and have responded to your comments point by point as follows:

Major Concerns:

1. Justification of Strain and Peptide Selection: While D-Q7 has been previously characterized and is a rational choice, the manuscript would benefit from a clearer explanation of why this particular peptide was selected to test against *A. hydrophila*, beyond citing past work. More importantly, the authors should discuss the broader antibiotic susceptibility profile of the colistin-resistant 23-C-23 strain. Since colistin resistance does not necessarily imply pan-resistance, it would help contextualize the clinical significance of D-Q7's activity - is D-Q7 filling an unmet need, or would standard antibiotics still work?

Thank you for your constructive comments. We fully agree with you on the need to explain more clearly in the paper why this particular peptide was chosen to test *Aeromonas hydrophila* rather than just citing previous studies, as well as to broaden the antibiotic susceptibility profile of strain 23-C-23, and we have added the appropriate content in the Introduction and Results sections, as described below.

1. Why D-Q7 was chosen for *A. hydrophila*

For stability, D-Q7 consists of all d type amino acids that is protease-resistant. For activity, from line 75 to line 78 in the revised manuscript, we add the sentence “23-C-23 was also a multi-drug-resistant *A. hydrophila* strain due to significantly high MICs of many conventional antibiotics against it, while the antibacterial activity of D-Q7 (MIC=4 µg/ml) against 23-C-23 is much more potent than colistin and other conventional antibiotics”

2. Expanded antibiotic-susceptibility profile of strain 23-C-23

We added the expanded antibiotic susceptibility profile of strain 23-C-23, New data added as **Table 1.** as follows:

MIC ($\mu\text{g/ml}$)	Antibiotics								
23-C-23	Penicillin	Lincomycin	Streptomycin	Kanamycin	Gentamicin	Tetracycline	Erythromycin	Azithromycin	Cefradine
	4096	512	4096	2048	16	8	16	32	512

Key finding: 23-C-23 is resistant to clinically relevant antibiotic, confirming a multidrug-resistant (MDR) phenotype in addition to colistin resistance. D-Q7 remained active at 4 $\mu\text{g/ml}$, indicating that it addresses an unmet therapeutic need. We believe these additions clarify the rationale for selecting D-Q7 and demonstrate that its potency fills a clinically relevant gap where conventional antibiotics are not effective. We appreciate your guidance in strengthening the manuscript.

2. Mechanistic Interpretation of gene3832: While gene3832 is convincingly shown to affect D-Q7 susceptibility, the claim that it represents a mechanism of resistance could be toned down or clarified. The study relies on transcriptomic association and knockout data, but additional mechanistic insight (e.g., functional characterization of the protein product) is missing. Suggesting it as a potential resistance modulator is appropriate, but stronger language should be reserved unless more direct mechanistic evidence is available.

We fully agree with your valuable comments, although in this study we have genetically (knockout/complementation) resolved gene3832 affecting D-Q7 susceptibility and its effect on bacterial membrane permeability without delving into its protein level or structure-function relationship. Follow-up studies will complement the investigation of gene 3832 at the protein level to more fully elucidate the mechanism of resistance to D-Q7. Thus, in the current study we have modified it as a potential resistance regulator. Thank you for the thoughtful feedback that helps us strengthen the manuscript.

3. Comparison to Other Antibiotics: The study compares D-Q7 to colistin, but not to other antibiotics (e.g., aminoglycosides, quinolones). Including a brief comparison or referencing prior susceptibility profiles would help readers understand how D-Q7 fits into the current therapeutic landscape.

We appreciate your suggestion to position D-Q7 within the broader antimicrobial spectrum. In the revised manuscript we have expanded the comparative analysis as follows:

MIC ($\mu\text{g/ml}$)	D-Q7									
23-C-23	4									
23-C-23:Δgene3832	1									
23-C-23:CΔgene3832	4									
23-C-23:ΔenvZ/ompR	0.5									
EGCT42	4									
RTF13	8									
WWO29	8									
23-C-23	Antibiotics									
	Penicillin	Lincomycin	Streptomycin	Kanamycin	Gentamicin	Tetracycline	Erythromycin	Azithromycin	Cefradine	
	4096	512	4096	2048	16	8	16	32	512	

These data confirm that strain 23-C-23 exhibits multidrug resistance (MDR) across β -lactams, aminoglycosides, and tetracyclines in addition to colistin; D-Q7 remains the only compound with retained activity (MIC = 4 $\mu\text{g/ml}$). We believe these additions clearly demonstrate how D-Q7 compares with—and in this MDR context surpasses—conventional antibiotics, thereby addressing your request. Thank you for helping us clarify the clinical relevance of our work.

Minor Suggestions:

1. The writing is generally clear, but would benefit from language polishing in

several areas to improve flow and precision. For example, phrasing like "the bacteria burden was determined" could be reworded for clarity.

We appreciate your careful reading and suggestions for improving linguistic clarity. In the revised manuscript we have conducted a comprehensive English-language edit, assisted by a native-speaking scientific editor, to enhance readability and precision. For example, specific revisions include the following: “the bacteria burden was determined” was modified to “we determine the bacterial load”.

2. Figure captions could be more descriptive and self-contained, especially for readers skimming the figures.

Thank you for your professional opinion! We have revised every figure caption so that each is now fully self-contained. Major improvements include:

“FIG 1 D-Q7 has significant sterilizing activity against colistin-resistant *A. hydrophila* 23-C-23. (A) 23-C-23 was treated with 2 µg/ml D-Q7 for 4 and 8 h, and then 23-C-23 cell shrinkage, cell wall damage, and organelle leakage were observed by transmission electron microscopy (TEM). (B-C) Treatment of WXC23 and 23-C-23 with 2 µg/ml D-Q7 or vehicle control (Ctr) resulted in an increase in bacterial reactive oxygen species (ROS) and a decrease in ATP. (D) Effects of exogenous addition of phosphatidylglycerol (PG), phosphatidylcholine (PC), and lipopolysaccharide (LPS) (0~200 µg/ml) on the anti-23-C-23 activity of D-Q7, respectively. (E) A murine epicutaneous infection model was established. 23-C-23 inoculation after D-Q7 treatment was set as prevention group; D-Q7 treatment after 23-C-23 inoculation was set as therapeutic group. (F) On day 5, the skin infected on mice was extracted and determining the bacterial load.”

“FIG 2 Gene3832 is involved in D-Q7 activity against 23-C-23 strain. (A-B) Transcriptome KEGG enrichment analysis of 23-C-23 strain treated with 2 µg/ml D-Q7 for 4 and 8 hours. (C) Heat map was shown for the transcriptional level change of 23-C-23 and WXC23 strains for specific genes with D-Q7 treatment for 4 or 8

hours. (D) Strain 23-C-23: Δ 3832, wild-type 23-C-23 and 23-C-23:C Δ 3832 exhibit marked alterations in membrane permeability. (E) Strain 23-C-23: Δ 3832, wild-type 23-C-23 and 23-C-23:C Δ 3832 undergo altered biofilm formation capacity after D-Q7 treatment. (F) Relative expression levels of *gene3832* in different *Aeromonas hydrophila* strains treated with D-Q7 as determined by quantitative RT-PCR.”

Thank you again for helping us improve the presentation quality.

3. A short, one-sentence rationale for selecting D-Q7 in the Introduction would help orient readers unfamiliar with the peptide.

Thank you for your insights! In the introduction we have made the following changes: from line 37 to line 40 in the revised manuscript, “**D-Q7 is a D-type antimicrobial peptide with potent sterilization activity against gram-negative ESKAPE pathogens, it is thus of considerable interest to evaluate whether D-Q7 represents a promising therapeutic candidate against this pathogen.**” We believe this concise statement will quickly orient readers who are not yet familiar with D-Q7.

Re: Spectrum00590-25R1 (**Killing of colistin-resistant *Aeromonas hydrophila* by a synthetic peptide**)

Dear Prof. Zhiliang Sun:

Your manuscript has been accepted, and I am forwarding it to the ASM production staff for publication. Your paper will first be checked to make sure all elements meet the technical requirements. ASM staff will contact you if anything needs to be revised before copyediting and production can begin. Otherwise, you will be notified when your proofs are ready to be viewed.

Sincerely,
Jinshui Lin
Editor
Microbiology Spectrum

Reviewer #1 (Comments for the Author):

The authors have addressed the major and minor concerns raised during the initial review. Their revisions significantly strengthen the manuscript's scientific rigor, clinical relevance, and clarity.